**Data Availability Statement:** All relevant data are within the paper and its Supporting Information files.

# Access to and challenges in water, sanitation, and hygiene in healthcare facilities during the early phase of the COVID-19 pandemic in Ethiopia: A mixed-methods evaluation

**Gete Berihun** [1]⊛*, **Metadel Adane** [1]⊛, **Zebader Walle** [2]⊛, **Masresha Abebe**[1], **Yeshiwork Alemnew**[3], **Tarikuwa Natnael**[1], **Atsedemariam Andualem**[4], **Sewunet Ademe**[4], **Belachew Tegegne**[4], **Daniel Teshome**[5], **Leykun Berhanu**[1]

1 Department of Environmental Health, College of Medicine and Health Sciences, Wollo University, Dessie, Ethiopia, 2 Department of Public Health, College of Health Sciences, Debre Tabor University, Debre Tabor, Ethiopia, 3 Department of Biology, College of Natural Sciences, Wollo University, Dessie, Ethiopia, 4 Department of Nursing, College of Medicine and Health Sciences, Wollo University, Dessie, Ethiopia, 5 Department of Anatomy, College of Medicine and Health Sciences, Wollo University, Dessie, Ethiopia

⊛ These authors contributed equally to this work.
* geteberihun@gmail.com

## Abstract

### Background

Inadequate water, sanitation, and hygiene (WASH) in healthcare facilities (HCFs) have an impact on the transmission of infectious diseases, including COVID-19 pandemic. But, there is limited data on the status of WASH facilities in the healthcare settings of Ethiopia. Therefore, this study aimed to assess WASH facilities and related challenges in the HCFs of Northeastern Ethiopia during the early phase of COVID-19 pandemic.

### Methods

An institution-based cross-sectional study was conducted from July to August 2020. About 70 HCFs were selected using a simple random sampling technique. We used a mixed approach of qualitative and quantitative study. The quantitative data were collected by an interviewer-administered structured questionnaire and observational checklist, whereas the qualitative data were collected using a key-informant interview from the head of HCFs, janitors, and WASH coordinator of the HCFs. The quantitative data were entered in EpiData version 4.6 and exported to Statistical Package for Social Sciences (SPSS) version 25.0 for data cleaning and analysis. The quantitative data on access to WASH facilities was reported using WHO ladder guidelines, which include no access, limited access, and basic access, whereas the qualitative data on challenges to WASH facilities were triangulated with the quantitative result.

### Results

From the survey of 70 HCFs, three-fourths 53 (75.7%) were clinics, 12 (17.2%) were health centers, and 5 (7.1%) were hospitals. Most (88.6%) of the HCFs had basic access to water

**Funding:** Wollo University funded this research project. The funders had no role in study design, data collection and analysis, decisions to publish, interpretation of the data, and preparation of the manuscript for publication.

**Competing interests:** The authors have declared that no competing interests exist.

**Abbreviations:** COVID 19, Coronavirus Disease 2019; HCFs, healthcare facilities; HCAI, healthcare-acquired infection; LMIC, low- and middle-income countries; PPE, personal protective equipment; WASH, water, hygiene, and sanitation.

supply. The absence of a specific budget for WASH facilities, non-functional water pipes, the absence of water-quality monitoring systems, and frequent water interruptions were the major problems with water supply, which occurred primarily in clinics and health centers. Due to the absence of separate latrine designated for disabled people, none of the HCFs possessed basic sanitary facilities. Half (51.5%) of the HCFs had limited access to sanitation facilities. The major problems were the absence of separate latrines for healthcare workers and clients, as well as female and male staffs, an unbalanced number of functional latrines for the number of clients, non-functional latrines, poor cleanliness and misuse of the latrine. Less than a quarter of the HCFs 15 (21.4%) had basic access to handwashing facilities, while half 35 (50%) of the HCFs did not. The lack of functional handwashing facilities at expected sites and misuse of the facilities around the latrine, including theft of supplies by visitors, were the two most serious problems with hygiene facilities.

## Conclusion

Despite the fact that the majority of HCFs had basic access to water, there were problems in their sanitation and handwashing facilities. The lack of physical infrastructure, poor quality of facilities, lack of separate budget to maintain WASH facilities, and inappropriate utilization of WASH facilities were the main problems in HCFs. Further investigation should be done to assess the enabling factors and constraints for the provision, use, and maintenance of WASH infrastructure at HCFs.

## Background

The provision of water, sanitation, and hygiene (WASH) facilities plays a crucial role in the reduction of healthcare-acquired infection (HCAI). Proper utilization of these facilities in healthcare settings is considered as a cornerstone for providing good quality care [1]. The common prevention measures against HCAI are source control, respiratory hygiene, early identification and isolation of patients with suspected disease, handwashing, and use of personal protective equipments (PPE) [2, 3].

The issue of adequate WASH is a regional problem in countries around the world, it is most severe in low-and middle-income countries (LMICs), including Ethiopia [4]. It is usually aggravated by the presence of a weak healthcare system and insufficient investment in healthcare safety. Hence, the implementation of proper infection prevention and control measures is challenging due to inadequate supplies of PPE. During the COVID-19 pandemic, adequate care of COVID-19 patients and prevention of HCAI among healthcare workers is difficult because of simple, but often neglected factors such as a lack of water [5]. Around 1.4 million people are affected by a lack of clean and safe healthcare facilities around the world. The problem is 2 to 20 times higher in low-resource countries than in developed countries [6].

A quarter of HCFs worldwide lack basic water services, exposing 1.8 billion people at risk, especially the most vulnerable groups of the population, such as healthcare workers and patients that attend HCFs. Furthermore, one-third of HCFs lack hand hygiene facilities at the point of care, and 10% of HCFs lack sanitation services. Globally, in 47 least-developed countries, an estimated half of HCFs do not have basic water services and two-thirds of HCFs lack basic sanitation services. Seven out of ten HCFs in least-developed countries do not have basic healthcare waste management services. About 50% of the HCFs in least-developed countries

had basic water services, 37% had basic sanitation, and 74% had basic hand hygiene facilities at the point of care [7].

Despite the improvement in access to essential health services in sub-Saharan Africa in recent years, the quality of care received remains inadequate to improve health outcomes. Health facilities lack the necessary infrastructure, equipment, medicines, commodities, and trained personnel to create an enabling environment, resulting in missed opportunities to provide good quality essential health services. About one-fifth of deaths occurring in LMICs are attributable to the lack of access to health services, one-third of deaths are a result of receiving poor quality of care, which is often linked to insufficient readiness of the facilities to provide services [8].

Access to WASH facilities in HCFs is a cornerstone of safe healthcare services [9]. The lack of these facilities poses significant health risks to patients, healthcare workers, and the whole community. WASH facilities in HCFs are fundamental to health security, preparedness, and response efforts, including the effort to stop the COVID-19 pandemic [7]. The lack of WASH facilities is one of the primary causes of the transmission of HCAI, including COVID-19 [10]. The Sustainable Development Goal (SDG) target 6 calls for universal access to WASH services in HCFs [11]. Globally, improving WASH facilities has the potential to prevent at least 9.1% of the disease burden in disability-adjusted life years or 6.3% of all deaths [12].

A study conducted in LMICs reported und that 38.0% of HCFs did not have a basic water supply, 19.0% did not have basic sanitation, and 35.0% did not have water and soap for handwashing [13]. The rate of provision of water is lowest in the African region, with 42.0% of all HCFs lacking an improved water source on-site or nearby [14]. Inadequate WASH in HCFs has a significant negative influence on the status of hospital patients' health during their stay. Globally, an estimated 15% of patients may acquire one or more infections during their stay in the hospital. But, the prevalence may be even higher in LMIC where it ranges from 5.7% to 19.1% and the risks associated with sepsis are 34 times higher [15].

Effective WASH plays a vital to prevent and control the transmission of COVID-19 [16]. The current COVID-19 pandemic has highlighted deficiencies in access to WASH services in HCFs and underscored the need for increased political commitment and enhanced accountability to address WASH gaps in health facilities [8]. According to the reports of WHO, confirmed cases of COVID-19 reached more than 228 million as of September 19, 2021, and caused more than 4.7 million deaths across the world [17]. The number of COVID-19 infections among healthcare workers is far greater than among the general population due to their role in treatment and management of cases [18, 19]. Globally, healthcare workers represent less than 3% of the population but account for 14% of COVID-19 cases [20]. The first case of COVID-19 in Ethiopia was reported on March 13, 2020 [21–24]. As of October 24, 2021, Ethiopia had reported a total of 362,088 COVID-19 confirmed cases and 6,347 deaths [25].

In May 2019, the World Health Assembly passed a resolution to accelerate global efforts on WASH in HCFs. This resolution led to a subsequent global meeting where countries presented their national commitments with concrete actions [26]. The government of Ethiopia has also implemented various COVID-19 prevention measures, such as partial or total lockdown, physical distancing, handwashing, and others [27–29].

Improving WASH in HCFs facilities is considered as a first-line defense against infectious disease [30]. But still there is no standard WASH guideline in HCFs of Ethiopia. To date, there is a lack of evidence on access to and challenges around WASH facilities in Ethiopian HCFs, including South Wollo Zone health facilities in Northeastern Ethiopia. Therefore, this study was designed to assess WASH facilities and related challenges in healthcare facilities of Northeastern Ethiopia in the early phase of the COVID-19 pandemic.

## Methods and materials

### Study design, period, and area

An institution-based cross-sectional study was conducted during July and August 2020 in 70 HCFs found in the South Wollo Zone, one of 15 Zones in the Amhara Region of Ethiopia. Based on the 2014 population projection, South Wollo had a total population of 2,925,559 of which 1,448,174 and 1,477,385 were male and female, respectively [31]. According to the Zonal Health Department report, South Wollo Zone has 7 governmental hospitals and 3 private hospitals, 135 health centers, 496 health posts, and 175 clinics [32].

### Source and study population

All HCFs that existed in the South Wollo Zone at the time of data collection were the source population. All randomly selected HCFs in South Wollo Zone were the study population.

### Sample size determination and sampling procedures

From the total number of HCFs, 70 HCFs were randomly selected using a lottery method from the lists of HCFs from the zonal health department of South Wollo. For the qualitative data collection, senior janitors, WASH coordinators of the HCFs, heads of HCFs, and clients from in-patient departments were purposively selected. A total of 14 participants, including 3 heads of HCFs, 3 WASH coordinators, 4 janitors, and 4 clients from in-patient departments were participated in the qualitative data collection.

### Operational definitions

**Healthcare facilities.** All formally recognized facilities that provide healthcare, including primary (health posts and clinics), secondary, and tertiary (district or national hospitals), public and private (including faith-run), and temporary structures designed for emergency contexts [33].

**WASH.** All works related to water, sanitation, and hygiene, including the provision of safe and affordable access to a clean water supply, sanitation, and hygiene service facilities [33].

**Improved water source.** Water sources from piped water, boreholes or tube wells, protected dug wells, protected springs, and rainwater [33].

**Improved sanitation.** Facilities that are designed to hygienically separate excreta from human contact, including flush/pour-flush to a piped sewer system, septic tank, or pit latrine; ventilated improved pit latrines, composting toilets, or pit latrines with slabs [33].

### Water supply

**Basic access water supply.** Water is available from an improved water source on the premises [34].

**Limited access water supply.** Improved water sources within 500 m of the premises but not all requirements for basic services are met [34].

**No water access.** Water is taken from an unprotected dug well or spring or surface water source or improved water source that is located more than 500 meters from the premises, or there is no source of water [34].

### Sanitation

**Basic access sanitation.** Improved sanitation facilities that are usable with at least one toilet dedicated for staff, at least one sex-separated toilet with menstrual hygiene facilities, and at least one toilet accessible to people with limited mobility [34].

**Limited access to sanitation.** At least one improved sanitation facility is available, but not all requirements for basic service are met [34].

**No sanitation access.** Toilet facilities are unimproved (e.g., pit latrines without a slab, or platform, hanging latrine, bucket latrine, or there is no latrine [34].

## Hygiene facilities

**Basic access hygiene facilities.** Functional handwashing facilities (with water and soap and/or an alcohol-based hand rub are available at the point of care and within five meters of toilets [34].

**Limited access hygiene facilities.** Functional hand hygiene facilities are available either at the point of care or near toilets, but not both [34].

**No hygiene facilities access.** No functional hand hygiene facilities are available either at the point of care or near toilets [34].

**Proper waste management.** Waste is safely segregated into at least three bins, and sharps and infectious waste are treated and disposed of safely [34].

**Low cleanliness latrine.** Visible faeces and/or urine observed on the floor around the latrine and latrine not swept at the time of data collection [35].

**High cleanliness latrine.** Pit not full, no faecal matter seen around the pit latrine, area properly swept, and absence of bad smell at the time of data collection [35].

## Data collection tools and quality assurance

A structured questionnaire that was adapted from WHO guidelines and published papers [34, 36–38] and contextualized based on the study setting. The questionnaire was prepared in English, translated to the local language (Amharic), and then re-translated to the English language for consistency. The questionnaire consisted of questions on general background information on the HFCs, water supply, sanitation facilities, hygiene facilities, and major challenges of the WASH facilities.

The quantitative data was collected using a combination of an interviewer-administered questionnaire and an observational checklist. For the qualitative data, we used key-informant interviews from purposively selected janitors, WASH coordinators of HCFs, heads of HCFs, and clients of the HCFs from in-patient departments. The data were collected by four professionals with Bachelor of Science (BSc.) in environmental health who had experience of working in WASH activities and supervision was conducted by two WASH experts.

Two days of training was given by the principal investigator for the data collectors and supervisors on the data collection procedures, the content of the tool, and ethical considerations. The questionnaire was pre-tested in 5% of the final sample size to assure the validity of the measuring tool; amendments were made based on the feedback from the pre-test including improving the order of questions, editing unclear questions, and eliminating less important questions. The collected data was checked daily and any missed data were collected immediately by re-visiting the health facility. Re-checking of the entered data was done using 10% of the sample size to control data entry errors and then data cleaning was done before statistical analysis.

## Data processing and analysis

The data were entered in EpiData version 4.6 and exported to Statistical Package for Social Sciences (SPSS) version 25.0 for data cleaning and analysis. For quantitative data, descriptive statistics were calculated for categorical variables and mean ± standard deviations (SD) for continuous variables. The access to WASH facilities at HCFs was categorized based on WHO

ladder guidelines in terms of basic, limited, or no access. The finding of the qualitative survey was triangulated with the access to WASH in HCFs findings to provide stronger evidence.

### Ethical considerations

Ethical clearance was obtained from the ethical review committee of Wollo University, College of Medicine and Health Sciences with ethical letter protocol number: CMHS/451/013/2020. Permission was obtained from South Wollo Zone Health Bureau and the respective HCFs. Before beginning data collection, the purpose of the study was explained to the study participants. Written consent was obtained from participants who could read and write whereas verbal/ oral consent was obtained from those who could not. The data collectors wore facemasks and maintained social distancing as per the WHO guidelines for the prevention of COVID-19. Facemasks were provided for study participants who did not wear them during the data collection period. The anonymity of the study participants was ensured by avoiding possible identifiers such as names. All the information obtained from the study participants was kept confidential.

## Results

### Background information of the HCFs

Of the total surveyed 70 HCFs, three-fourths 53 (75.7%) of them were clinics, 12 (17.2%) were health centers, and the remaining 5 (7.1%) were hospitals. More than three-fourths 57 (81.4%) of the HCFs employed Environmental Health professionals who were responsible for coordinating WASH facilities in the healthcare setting. The mean daily client flow rate was 55±108 (Table 1).

The head of HCF reported that "the major problem of WASH in our healthcare facility is the absence of environmental health professionals to monitor WASH facilities and unorganized WASH committee."

### Access and challenges water supply facilities in HCFs

Regarding water supply, all investigated HCFs used tap water as the main source of water. Eight (11.4%) of HCFs did not have water during the time of data collection. One-third 27 (38.6%) of the HCFs had water storage containers that could be used as a reservoir during interruptions of the main water supply. The average number of taps in hospitals, health centers, and clinics were 80, 10, and 5, respectively. More than three-quarters of the HCFs 56

**Table 1. Background information of the healthcare facilities in Northeastern Ethiopia from July to August 2020.**

| Variable | Category | Type of healthcare facility (N = 70) | | | Frequency (n) | Percentage (%) |
|---|---|---|---|---|---|---|
| | | Hospital (n = 5) | HC (n = 12) | Clinic (n = 53) | | |
| Ownership | Private | 3 | 0 | 53 | 56 | 80 |
| | Government | 2 | 12 | 0 | 14 | 20 |
| Location of HCF | Urban | 5 | 11 | 50 | 66 | 94 |
| | Rural | 0 | 1 | 3 | 4 | 6 |
| Presence of WASH coordinator | Yes | 2 | 11 | 0 | 13 | 19 |
| | No | 3 | 1 | 53 | 57 | 81 |
| Presence of WASH committee | Yes | 3 | 8 | 3 | 14 | 20 |
| | No | 2 | 4 | 50 | 56 | 80 |
| The average number of employees | Mean ±SD | 192±122 | 32±4 | 10±2 | 27±57 | |
| The average number of patients | Mean± SD | 319±212 | 90±40 | 18±6 | 55±108 | |

**Table 2. Water supply facilities in healthcare facilities of Northeastern Ethiopia from July to August 2020.**

| Variable | Category | Type of healthcare facility (N = 70) | | | Total | |
|---|---|---|---|---|---|---|
| | | Hospital (n = 5) | HC (n = 12) | Clinic (n = 53 | Frequency (n) | Percentage (%) |
| Presence of alternate water storage container | Yes | 5 | 10 | 12 | 27 | 39 |
| | No | 0 | 2 | 41 | 43 | 61 |
| Time to fetch water in minutes | <5 | 0 | 0 | 22 | 22 | 31 |
| | 5–10 | 5 | 6 | 23 | 34 | 49 |
| | 10–15 | 0 | 6 | 3 | 9 | 13 |
| | >15 | 0 | 0 | 5 | 5 | 7 |
| Presence of water during survey | Yes | 5 | 12 | 45 | 62 | 89 |
| | No | 0 | 0 | 8 | 8 | 11 |
| Presence of water during the last two weeks | Yes | 5 | 11 | 48 | 64 | 91 |
| | No | 0 | 1 | 5 | 6 | 9 |
| Water supply present on an annual basis | Yes all year | 5 | 9 | 21 | 35 | 50 |
| | Most of the time | 0 | 3 | 30 | 33 | 47 |
| | Not present mostly | 0 | 0 | 2 | 2 | 3 |
| Presence of water quality monitoring program | Yes | 4 | 2 | 8 | 14 | 20 |
| | No | 1 | 10 | 45 | 56 | 80 |
| Average number of taps available | | 80 | 10 | 5 | | |
| Average number of functional taps | | 72 | 8 | 4 | | |

(80%) did not have a system for frequent water quality monitoring in their healthcare setting. Finally, 62 (88.6%) of the HCFs had basic access to a water supply (Table 2).

The head of HCFs said that: "the major problem regarding the water supply was the absence of a separate budget for WASH facilities such as repairing of the damaged pipes."

Clients from an inpatient department said that "we do not use water from healthcare facilities especially at night because of its distance from the in-patient ward; as a result, we are obligated to use other sources of water such as bottled water, which exposes us to extra cost."

## Access to and challenges around sanitation facilities in HCFs

About half 36 (51.5%) of the HCFs used improved sanitation facilities. This study also revealed that slightly more than half 39 (55.7%) of HCFs had separate toilets for male and female while one-third 23 (32.9%) had separate toilets for clients and workers. None of the HCFs had a latrine designed for disabled people. More than three-quarters 57 (81.4%) of the HCFs reported that there was an insufficient supply of PPE for healthcare workers in their healthcare setting (Table 3).

The focal persons of WASH said that "the major problem with sanitation in the HCFs was unbalanced patient load with the availability of a functional latrine on selected dates Monday, Friday, and market days, particularly in the morning session."

An in-patient client of HCF said that "the major problem of sanitation in the HCF was the non-functionality of latrines (broken doors, locked latrines), absence of separate toilets to take a sample, and lack of cleanliness of the surface of the latrine."

A janitor of the healthcare facility said that "the existing challenges regarding sanitation are that although we clean the slab of latrine frequently, there are great problems in proper disposal of faeces and urine, particularly during sample collection by clients for laboratory examination."

**Table 3. Sanitation facilities based on the type of healthcare facility in Northeastern Ethiopia from July to August 2020.**

| Variable | Category | Types of healthcare facility | | | Total | |
|---|---|---|---|---|---|---|
| | | Hospital (*n*) | Health center (*n*) | Clinic (*n*) | Frequency (*n*) | Percentage (%) |
| Type of latrine | Flush | 0 | 1 | 1 | 2 | 3 |
| | VIP | 5 | 4 | 23 | 32 | 46 |
| | Pit latrine with slab | 0 | 1 | 1 | 2 | 3 |
| | Pit latrine without slab | 0 | 6 | 28 | 34 | 49 |
| Presence of functional doors and locks in toilets | Yes | 5 | 6 | 27 | 38 | 54 |
| | No | 0 | 6 | 26 | 32 | 46 |
| Absence of odor | Yes | 5 | 7 | 28 | 40 | 57 |
| | No | 0 | 5 | 25 | 30 | 43 |
| Separate toilet for male and female staff | Yes | 5 | 11 | 23 | 39 | 56 |
| | No | 0 | 1 | 30 | 31 | 44 |
| Separate toilets for patients and workers | Yes | 5 | 7 | 11 | 23 | 33 |
| | No | 0 | 5 | 42 | 47 | 67 |
| Number of toilets | Mean | 20.4 | 5.08 | 3 | 323 | 5 |
| Patient to latrine ratio | | 16 | 18 | 6 | 3635 | 12 |
| Frequency of toilet cleaning | Twice a day | 0 | 2 | 2 | 4 | 6 |
| | Three times a day | 0 | 4 | 23 | 27 | 39 |
| | More than three times a day | 5 | 6 | 28 | 39 | 56 |
| Do janitors use PPE while cleaning? | Yes | 5 | 12 | 40 | 57 | 81 |
| | No | 0 | 0 | 13 | 13 | 19 |
| Overall latrine cleanliness | High | 5 | 8 | 12 | 25 | 36 |
| | Medium | 0 | 3 | 37 | 40 | 57 |
| | Low | 0 | 1 | 4 | 5 | 7 |
| Excreta disposal method when latrine is filled | Emptying | 5 | 11 | 49 | 65 | 93 |
| | Covering with soil | 0 | 1 | 4 | 5 | 7 |

## Access to and challenges around hygienic facilities in HCFs

In this study, half 35 (50%) of HCFs had no access to handwashing facilities and 28 (40%) had functional sinks. Less than a quarter 15 (21.4%) of the HCFs had handwashing facilities (soap and water) at the point of care and near latrines. The study also showed that only 28 (40%) of HCFs practiced proper segregation of waste using three color-coded collection containers. On the contrary, 42 (60%) of HCFs (mainly clinics and health centers) had no proper waste management system (Table 4).

A client from a private hospital said that "the major WASH problem was the owner of the healthcare facility mainly focuses on owner's benefit rather than providing necessary facilities."

A client from a health center said that "there were no sufficient supplies of handwashing materials (water, soap or alcohol-based hand rub), mainly around latrines."

A WASH coordinator said that "Although we put out handwashing materials (water, soap and alcohol-based hand rub), there is a behavioral problem in the utilization of the facilities and some clients may even steal from these facilities, particularly at latrines."

## Discussion

Lack of access to WASH services hampers the implementation of preventive measures against SARS-CoV-2 and causes high mortality from diseases caused by diarrhea and lower respiratory infections [39]. Therefore, ensuring good and consistent WASH facilities in all settings, particularly in HCFs helps to prevent transmission of the SARS-CoV-2 virus.

**Table 4. Hygiene facilities based on the type of healthcare facility in Northeastern Ethiopia from July to August 2020.**

| Variable | Category | Types of healthcare facility (N = 70) | | | Frequency (n) | Percentage (%) |
|---|---|---|---|---|---|---|
| | | Hospital (n = 5) | HC (n = 12) | Clinic (n = 53) | | |
| Presence of functional sinks | Yes | 4 | 8 | 16 | 28 | 40 |
| | No | 1 | 4 | 37 | 42 | 60 |
| Presence of handwashing facility (soap, water/ABHR) at point of care and toilet | Yes | 4 | 5 | 6 | 15 | 21 |
| | No | 1 | 7 | 47 | 55 | 79 |
| Presence of handwashing facility (water, soap/ABHR) at point of care | Yes | 4 | 8 | 14 | 26 | 37 |
| | No | 1 | 4 | 39 | 44 | 63 |
| Presence of handwashing facility (water, soap/ABHR) at toilets | Yes | 4 | 5 | 12 | 21 | 30 |
| | No | 1 | 7 | 41 | 49 | 70 |
| Presence of handwashing poster about COVID-19 | Yes | 5 | 10 | 46 | 61 | 87 |
| | No | 0 | 2 | 7 | 9 | 13 |
| Segregation of types of waste into three bins accordingly | Yes | 4 | 7 | 17 | 28 | 40 |
| | No | 1 | 5 | 36 | 42 | 60 |

ABHR = Alcohol-based hand rub.

A safe and adequate water supply in healthcare facilities is vital for reducing the transmission of infectious diseases, including the current pandemic of COVID-19. HCFs require adequate quantity and quality of water to maintain a safe environment [40]. This study revealed that most 62 (88.6%) of the HCFs had basic access to a water supply, which was higher than the studies conducted in African countries (71.2%) [41], Ethiopia (30%), Uganda (44%), Tanzania (56%), Somalia (67%), and Rwanda (73%) [7], and Uganda (86%) [14].

The major problems regarding the water supply were interruption of the water supply system, absence of a water quality monitoring system, and a lack of a separate budget for the WASH services. The finding also revealed that no HCFs had any plan for WASH risk assessment maintenance. The lack of planning may be due to the absence of a separate budget for WASH service at HCFs mainly in LMICs which reported between 0.08 and 2.54% of Gross Domestic Product (GDP) invested in WASH. The major sources of funding are official development assistance, foundations and charities, and loans from international sources, which accounted for 12% of total finance in 2016–2018 [42].

Regarding sanitation, only half (51.5%) of the HCFs had improved sanitation facilities, which was lower than the studies conducted in African countries, such as Ethiopia (66%), Kenya (86%), Mozambique (79%), Rwanda (93%), Uganda (93%) and Zambia (96%) [41], LMIC (67%) [12], Rwanda (44%) [40], sub-Saharan Africa (94.3%) [8], LMIC (81%) [39], Ethiopia (76%), Rwanda (99%), Djibouti (95%) and Uganda (75%) [7], and Zimbabwe (98%) [28]. The average ratio of latrines to clients for HCFs were 1:12 which was lower than the WHO guideline of 1:20 [7]. Clinics had the highest ratio (1:6), followed by health centers (1:16), and hospitals (1:18), which was consistent with a study conducted in Uganda [13]. In low resource settings, including Ethiopia, sanitation services in HCFs are of low priority and are often neglected [43].

Although all the HCFs had access to sanitation services, none of them had basic sanitation services, which was mainly attributed to the absence of latrines designed for disabled people. Half (51.5%) of the HCFs had limited access to sanitation services, which was lower than the finding in Uganda (84.5%) but higher than the finding in Ethiopia (17%) [7]. The possible reason for this variation may be the change in the study period and government commitment towards improving WASH facilities in healthcare settings.

WHO recommends people with suspected or confirmed SARS-CoV-2 should be provided a separate toilet, and if not possible, certain toilets should be designated as solely for the shared use of COVID-19 patients and not used by non-COVID-19 patients [20]. In this study, 56% of the HCFs had separate toilets for males and females, which was lower than the study finding in Ethiopia 94.3% [44]. This finding also revealed that 7.1% of HCFs had low cleanliness of the latrine, which may be associated with the absence of a functional lock on the latrine door and lack of lighting, frequency of cleaning, and the behavior of clients in utilizing of latrines. A systematic review conducted in LMIC revealed that the lack of cleanliness of toilets was a bigger problem than the absence of toilets, and that was attributed to a lack of safe and adequate access to water [43].

The presence of handwashing facilities with clean water and soap is a key preventive measure against the transmission of infectious diseases, including COVID-19 [27]. According to the WHO standard, functional hand hygiene facilities should be available at all critical points of the HCFs [7]. Less than a quarter (21.4%) of the HCFs had functional handwashing facilities (with water and soap) both in latrines and at point of care, which was lower than the studies conducted in Africa (28%) [41], Zimbabwe (30%) (26), LMICs [12], sub-Saharan Africa (67%) [8], Rwanda (32%) [40], Nigeria (66%), Rwanda (65%), Zimbabwe (58%) [7] and Ethiopia (74.28%) [44] but higher than Niger (4%) [7].

Only half (50%) of the HCFs had basic access to handwashing facilities, which was lower than the studies conducted in Uganda (56.9%) [45], sub-Saharan Africa (74%) [8], and Nigeria (85%) [46]. On the other hand, the current finding was higher than a study conducted in LMICs (22%) [40]. This low achievement of hygiene facilities may be due to poor access to water supply and misconduct of clients in the utilization of these facilities. Hence, the lack of fundamental hygiene facilities in HCFs may affect the quality of service given and may create a suitable environment for the transmission of HAI, including COVID-19 [12, 47].

One-third (37.1%) of the HCFs, soap was observed at the point of care, which was in line with a study conducted in Rwanda (33%) [40]. On the other hand, it was lower than the studies conducted in LMICs (61%) [13], Uganda (75%), Rwanda (70%), Burundi (66%), Ethiopia (65%), and Somalia (58%), but higher than Djibouti (35%) [7]. On the other hand, heads and WASH coordinators of HCFs reported that they had put sufficient soap near the latrine and at the point of care, mainly since the occurrence of the COVID-19 pandemic in Ethiopia. But, they mentioned that there is misuse and/ theft of handwashing facilities, which was supported by a study conducted in Rwanda [40]. Therefore, the provision of adequate hand hygiene facilities requires not only the presence of access to washing materials (water and soap) but also appropriate behaviors [42, 48].

## Limitation of the study

This study has certain limitations. Due to the nature of the data, which was taken from a small sample size, we were unable to carry out statistical analysis. The health posts were excluded from the study due to their insignificant roles in the treatment and management of COVID-19. The other limitation of the study is that the exact number of HCFs with basic access to the water supply may be lower than the current finding due to the lack of laboratory examination of water quality assessment. Furthermore, the use of a cross-sectional study design, which cannot show the causality of the study, was also a limitation of the study.

## Conclusions

Although most of the HCFs had basic access to a water supply, there were frequent interruptions of water, absence of water quality monitoring system, absence of a separate budget for

WASH services, and non-functional water sources. About half of the HCFs had limited access to sanitation facilities. The major problems were the absence of accessible latrines for disabled people, lack of separate latrines for healthcare workers and clients, as well as female and male staff, unbalanced numbers of functional latrines with number of clients, non-functional latrines, and misuse of the latrine mainly during sample taking for laboratory investigation. Less than a quarter of the HCFs had basic access to handwashing facilities at both the point of care and near the latrine.

The major problems were the lack of functional handwashing facilities at the critical points and misuse, particularly around the latrine. Therefore, to reduce the risk of HAIs, including COVID-19, immediate actions should be taken by the concerned governmental and non-governmental organizations to provide sufficient water for all users, disability-friendly sanitation facilities, and handwashing facilities. The issue of WASH should be encouraged in government planning and budgeting. The technical and logistical capabilities of health and safety committees should be strengthened in order to prevent the spread of coronavirus through education and awareness campaigns. Further investigation should be done to assess the enabling factors and constraints for the provision, use, and maintenance of WASH infrastructure at HCFs.

## Supporting information

**S1 File. Annex I English version questionnaires.**
(DOCX)

**S2 File. Annex II Amharic version questionnaires.**
(DOCX)

**S1 Dataset.**
(XLSX)

## Acknowledgments

We acknowledge South Wollo Zone Health Bureau for providing all the necessary information when needed. Our thanks also extended to the studied healthcare facilities and heads of healthcare facilities in the South Wollo Zone. We also thanks data collectors, supervisors, and study participants for their valuable cooperation during the data collection.

## Author Contributions

**Conceptualization:** Gete Berihun, Metadel Adane, Zebader Walle, Masresha Abebe, Yeshiwork Alemnew, Tarikuwa Natnael, Atsedemariam Andualem, Sewunet Ademe, Belachew Tegegne, Daniel Teshome, Leykun Berhanu.

**Data curation:** Gete Berihun, Metadel Adane, Zebader Walle, Masresha Abebe, Yeshiwork Alemnew, Tarikuwa Natnael, Atsedemariam Andualem, Sewunet Ademe, Belachew Tegegne, Daniel Teshome, Leykun Berhanu.

**Formal analysis:** Gete Berihun, Metadel Adane, Zebader Walle, Masresha Abebe.

**Funding acquisition:** Gete Berihun, Masresha Abebe, Yeshiwork Alemnew, Tarikuwa Natnael, Atsedemariam Andualem, Sewunet Ademe, Belachew Tegegne.

**Investigation:** Gete Berihun, Metadel Adane, Zebader Walle, Masresha Abebe, Yeshiwork Alemnew, Tarikuwa Natnael, Atsedemariam Andualem, Sewunet Ademe, Belachew Tegegne, Daniel Teshome, Leykun Berhanu.

**Methodology:** Gete Berihun, Metadel Adane, Zebader Walle, Masresha Abebe, Yeshiwork Alemnew, Tarikuwa Natnael, Atsedemariam Andualem, Sewunet Ademe, Belachew Tegegne, Daniel Teshome, Leykun Berhanu.

**Project administration:** Gete Berihun, Metadel Adane, Zebader Walle, Yeshiwork Alemnew, Tarikuwa Natnael.

**Resources:** Gete Berihun, Metadel Adane, Zebader Walle, Tarikuwa Natnael.

**Software:** Gete Berihun, Metadel Adane, Zebader Walle, Masresha Abebe, Atsedemariam Andualem, Sewunet Ademe, Belachew Tegegne, Daniel Teshome, Leykun Berhanu.

**Supervision:** Gete Berihun, Metadel Adane, Zebader Walle, Masresha Abebe, Yeshiwork Alemnew, Atsedemariam Andualem, Sewunet Ademe, Belachew Tegegne, Daniel Teshome, Leykun Berhanu.

**Validation:** Gete Berihun, Metadel Adane, Zebader Walle, Masresha Abebe, Yeshiwork Alemnew, Tarikuwa Natnael, Atsedemariam Andualem, Sewunet Ademe, Belachew Tegegne, Daniel Teshome, Leykun Berhanu.

**Visualization:** Gete Berihun, Metadel Adane, Zebader Walle, Yeshiwork Alemnew, Tarikuwa Natnael, Atsedemariam Andualem, Sewunet Ademe, Belachew Tegegne, Daniel Teshome.

**Writing – original draft:** Gete Berihun, Metadel Adane, Zebader Walle.

**Writing – review & editing:** Gete Berihun, Metadel Adane, Zebader Walle.

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
