## [Decision Letter · Decision Letter 0]

1 Dec 2021

PONE-D-21-35346Assessment of water, sanitation, and hygiene facilities access and challenges in Healthcare Facilities of Northeastern Ethiopia in COVID-19 Era: A mixed methods evaluationPLOS ONE

Dear Dr. Berihun,

Thank you for submitting your manuscript to PLOS ONE. After careful consideration, we feel that it has merit but does not fully meet PLOS ONE’s publication criteria as it currently stands. Therefore, we invite you to submit a revised version of the manuscript that addresses the points raised during the review process.

Please include the following items when submitting your revised manuscript:A rebuttal letter that responds to each point raised by the academic editor and reviewer(s). You should upload this letter as a separate file labeled 'Response to Reviewers'.A marked-up copy of your manuscript that highlights changes made to the original version. You should upload this as a separate file labeled 'Revised Manuscript with Track Changes'.An unmarked version of your revised paper without tracked changes. You should upload this as a separate file labeled 'Manuscript'.

We look forward to receiving your revised manuscript.

Kind regards,

Alison Parker

Academic Editor

PLOS ONE

Journal Requirements:

(NO authors have competing interests)

Reviewers' comments:

Reviewer's Responses to Questions

**Comments to the Author**

1. Is the manuscript technically sound, and do the data support the conclusions?

Reviewer #1: No

Reviewer #2: Yes

2. Has the statistical analysis been performed appropriately and rigorously? 

Reviewer #1: No

Reviewer #2: Yes

3. Have the authors made all data underlying the findings in their manuscript fully available?

Reviewer #1: No

Reviewer #2: Yes

4. Is the manuscript presented in an intelligible fashion and written in standard English?

Reviewer #1: No

Reviewer #2: Yes

5. Review Comments to the Author

Reviewer #1: This paper looks at water, sanitation and hygiene access during the period of COVID-19 in Ethiopia.

1.) The name of the institutional review board that approved the study is included only. Please add the approval number and indicate the form of consent obtained (written/oral). The research does not meet all applicable standards for the ethics of experimentation and research integrity.

2.) Data Availability states “Yes - all data are fully available without restriction.” Please include the full dataset as a supplement to this paper or include a link where it can be downloaded. The article does not adher to appropriate reporting guidelines and community standards for data availability.

3.) Line 69. Reference is missing to the WHO/UNICEF WASH in healthcare facilities (https://washdata.org/sites/default/files/2020-12/WHO-UNICEF-2020-wash-in-hcf.pdf)

4.) Line 69 to 73, this data is out of date. Please update and use the most recent 2020 WHO/UNICEF report.

5.) This statement is false “HCAI including the current pandemic of COVID-19 has occurred due to a lack of WASH facilities and improper utilization of it (9)” COVID-19 is a respiratory disease. Correct this statement.

6.) Line 80 to 82, this data is out of date. Please update and use the most recent 2020 WHO/UNICEF report.

7.) Line 96, “October 29” What year?

8.) Line 98. The date of the first case of COVID-19 globally is incorrect. It was in December 2019.

9.) Line 105. There is data on Ethiopia and COVID-19 in the WHO/UNICEF report. “In Ethiopia, a large assessment of facilities carried out as part of the COVID-19 response resulted in the mobilization of US$ 5 million to support IPC and WASH activities in 74 high-load hospitals. WHO and UNICEF launched the ‘Hand Hygiene for All’ (HH4A) global initiative in June 2020. It is a call to action for all of society to achieve universal hand hygiene and to stop the spread of COVID-19”

10.) Line 127, what is tottery?

11.) Was distance to water sources measured with a tape measure?

12.) Statistics, and other analyses are not performed to a high technical standard and are not described in sufficient detail. This is listed as a limitation later, but needs to be done prior to acceptance of the paper.

13.) The N=70, there is not a large enough dataset to have results to 0.1%. Please round all results presented to the nearest whole number.

14.) The study needs to compare to the recent Ethiopia WASH in healthcare facility work during COVID-19, this is a major gap of this manuscript https://washdata.org/sites/default/files/2020-12/WHO-UNICEF-2020-wash-in-hcf.pdf

15.) The study presents the results of original research, but does not compare to other recent studies in Ethiopia or other low- and middle-income countries during COVID-19.

16.) Conclusions presented are not supported by the data. Expand the conclusions. How could the findings of this work be implemented for better access, not just in Ethiopia but lessons learned for other low- and middle-income countries? At least give some practical options for solutions in a bulleted list. What can be done within resource limited environments? How much financial investment is needed at a minimum to improve conditions? Also, I missed seeing what further research you can suggest in this field.

17.) Finally, when you submit the corrected version, please do check thoroughly, in order to avoid grammar, syntax or structure/presentation flaws - please seek for professional English proofreading services or ask a native English-speaking colleague of yours in order to refine and improve the English in your paper.

Other comments:

1.) healthcare facilities should not be in capital in the title, or throughout the manuscript. Only proper nouns should be in capital letters.

Reviewer #2: Abstract

Sampling why not you use proportional allocation to the number of health care facilities since all health facilities are known by the regional health bureau.

Line 33 add the word “by” before the word “using “ and add “interviewer administered” before “structured questioner”

Line 50 who are nongovernmental health facilities? Change the word either NGO or partners

Background

line 98 it doesn’t state in which country specifically that CoVID-19 is occurred on March 13, Hence re-write this sentence.

Line99-100 the sentence is not complete.

Line 101 278 HCFs are in nationwide, or regionally or zonal level???????

Study design

Remove line 113,114,115 which talks about the boarder of Wello zone that much is not necessary

Source and study population

Line 121 All HCFs which is present…… change the word present by the word “exist”

Sample size determination and sampling procedures

Line 125 why not you mention the number of the total health facilities found in the zone to see whether the sample (70) is representative or not? And what was your base to take 70? Why not 80, 90,100 or above?

Out of the total sample, 80% are private clinics what was your base to take larger sample from private clinics.

Line 125 Since there is enough data on the number of health institutions found in the zonal health department, why don’t you use proportional sampling technique rather than simple random to make it more uniform and representative of each health facility category?

Line 127-131 how qualitative data were collected? Is it in-depth interview or FGDs? Explain well which method is best in exploring the challenges of an issue?

Line 150 what about the definition of limited access for sanitation??

Data collection tools and quality assurance

Line 165 contextualized to the study……

Line 170 you use spot observation to collect data but you didn’t mention the result that you obtain through this method in your result part.

Line 188 how the qualitative data were analyzed? It is not exhaustively written. Hence you should mention the method that used to analyze it.

Line 204-206; do you think that your result represent the HCFs that are found in the zone?

Line 206 are an environmental health professionals the only professionals who coordinates the WaSH activities? Did you ask the presence of even other professionals in the position?

Line 207 average is not the appropriate measures rather you use mean +/- SD since the average is 55 but the highest value is 320, do you see the gap?

Line 208; though the highest number of patient flow is in the hospital, you included in your study only 7% of them. Do you think your sample is representative?

Line 234 I think it is better to write “…HCFs was unbalanced patient load with functional…. ”

Line 240 it is better to change the word “the floor” by the word “the slab”…..disposal of “faces” by “faeces “ after that add the word “and” before urine

Line 242 ….challenges of hygienic facilities

Line 243 since it is 35 or half of your sample, you can’t say more than half of…..

Line 248 “proper waste management” what does it mean in this research context. It needs operational definition for this research

Line 248 …..and they disposed off …. Add one “f”

Line 249-252; the sentence is not clear. Is it about cleaning or cleaning protocol or the janitors training status?

Line 253-255 I didn’t understand about your sample the person from private health care facilities customer? If it is so remove it since they are two independent setting i.e governmental and private hospitals you compared two different institutions. Besides, in your sampling section, there is no list of participant from private health institution users. From where did you bring this information?

Line 262-263 it is better to add references to the written sentence

Line 265-267 the sentence which describes the aim should be omitted since it is already mentioned earlier

Line 267. Can we say all HCFs have accessed to water supply? In you result part line 214, 11% of them have no water during data collection time. Some time it is more than that. Hence you should modify this statement

Line 267-271 you should treat each component independently otherwise there will be writing the result repeatedly which will be boring to the reader. Hence write the discussion separately for water, sanitation and hygiene

Line 273 and 274 it needs re-writing and is it possible to put the different country status by average number? Why not you mention for each country status to know and compare it

Line 275; your justification doesn’t convince the reader i.e why don’t you find the similar setting to compare it either rural or urban?

Line 278 your justification is the variation in the HCFs why not you compare with similar setting?

Line 278 remove the bracket at the end of line 278

Line 279-282 “almost all …….in private clinics” this statement should be placed after discussing water, sanitation and hygiene results and even compare the result and the situation with studies done in other parts of Ethiopia.

Line 287 you said that the ratio 11.5 is greater than WHO standard. What is the WHO standard put the number. Besides, there is no reference, hence put your reference

Line 288 the statement “….between the level….” Should be changed to between the HCFs. Even it is better to put the ratio result (in number) with in each HCF.

Line 291-301 the paragraph seems the result part since you didn’t discuss any of your result by comparing your result with the other studies. Hence this paragraph needs re-writing again

Line 304 “HCFs had no function…..” change the word function to “functional”

Line 305 what about the other study setting you didn’t discuss your finding simply put the number that you wrote in the result part here again.

Line 307-309; this justification is for the 60% (HCFS had no function hand washing facilities) or 21% (HCFs had functional handwashing facilities with water and soaps)? It is confusing discussion part. Hence it is better to treat independently.

Line 314-320 you simply put your result again that you wrote in your result part. Hence it needs further discussion by comparing your result with other findings and give your justification why the variation occurs.

Where is the qualitative data discussion part? you didn’t put anything. Why you collect the qualitative data? And how you analyzed it? Is it by open code, thematic, ATLAS-Ti or by what technique? Nothing is said about this. Please write something on it. You said in your methodology mixed where is the mixed nature? I didn’t see it or realized it.

Line 323 “exclusion of health post” I preferred you were included these health post rather than the private clinics that you included. The reason is that

1. There is a mix up of different setting that is governmental and private which are almost completely different setting in Ethiopian condition

2. Your study design is not comparative cross sectional method

3. You didn’t compare the result of private clinics with the government or even to the other similar setting of different countries. The data that you got from private clinics are confined in the governmental HCFs and treated as they are governmental HCFs

Limitation part what about the cross-sectional nature of the data?

Line 335 make correct on the word “…..concerned exerts……” change to concerned experts

Annex

Table 3 on page 24 “overall latrine cleanliness” i.e high, medium, low, it needs operational definition.

6. PLOS authors have the option to publish the peer review history of their article (what does this mean?). If published, this will include your full peer review and any attached files.

Reviewer #1: No

Reviewer #2: **Yes: **Mathewos Moges

---

## [Author Response · Author response to Decision Letter 0]

15 Feb 2022

Response to editor

Question #1 Please ensure that your manuscript meets PLOS ONE's style requirements, including those for file naming.

Response: Thank you for this remark. We re-formatted the revised manuscript using the PLoS ONE format guidelines. The whole content of the manuscript, including the abstract, introduction, methods, discussion and reference are formatted using the guidelines (Please see the revised version for each section).

Question #2 Please complete your Competing Interests on the online submission form to state any Competing Interests. If you have no competing interests, please state "The authors have declared that no competing interests exist.” as detailed online in our guide for authors at http://journals.plos.org/plosone/s/submit-now. This information should be included in your cover letter; we will change the online submission form on your behalf

Response: thank you for your remark; we have incorporated it in the cover letter.

Question #3. Data availability

Response. We have attached the data on the supplementary information 

Response to Reviewer 1

Question ##1) The name of the institutional review board that approved the study is included only. Please add the approval number and indicate the form of consent obtained (written/oral). The research does not meet all applicable standards for the ethics of experimentation and research integrity.

Response: We have added the approval number and the form of consent in the revised manuscript. 

Question #2.) Data Availability states “Yes - all data are fully available without restriction.” Please include the full dataset as a supplement to this paper or include a link where it can be downloaded. The article does not adhere to appropriate reporting guidelines and community standards for data availability.

Response: We have included all necessary supplementary data in the revised version manuscript.

Question #3.) Line 69. Reference is missing to the WHO/UNICEF WASH in healthcare facilities (https://washdata.org/sites/default/files/2020-12/WHO-UNICEF-2020-wash-in-hcf.pdf)

Response: we have included the reference in the revised manuscript. 

Question #4.) Line 69 to 73, this data is out of date. Please update and use the most recent 2020 WHO/UNICEF report.

Response: thank you for your comment. We have used the most recent WHO/UNICEF 2020 report in the revised manuscript. 

Question #5.) This statement is false “HCAI including the current pandemic of COVID-19 has occurred due to a lack of WASH facilities and improper utilization of it (9)” COVID-19 is a respiratory disease. Correct this statement.

Response: sorry for the confusion we have created. Hence, we have re-written the sentence in the revised manuscript. Question #6.) Line 80 to 82, this data is out of date. Please update and use the most recent 2020 WHO/UNICEF report.

 Response: thank you for your comment. Hence, we have used the most recent 2020 WHO/UNICEF report in the revised manuscript.

Question #7 Line 96, “October 29” What year?

 Response: Sorry for the problem we have created. Hence, we have incorporated the missed data in the revised version of the manuscript. 

Question #8.) Line 98, the date of the first case of COVID-19 globally is incorrect. It was in December 2019.

Response: Sorry for the confusion we have created in the original manuscript. The idea of the sentence was the first case of COVID-19 in Ethiopia, not at the global level. Hence, we have rephrased the sentence accordingly in the revised manuscript. 

Question #9) Line 105. There is data on Ethiopia and COVID-19 in the WHO/UNICEF report. “In Ethiopia, a large assessment of facilities carried out as part of the COVID-19 response resulted in the mobilization of US$ 5 million to support IPC and WASH activities in 74 high-load hospitals. WHO and UNICEF launched the ‘Hand Hygiene for All’ (HH4A) global initiative in June 2020. It is a call to action for all of society to achieve universal hand hygiene and to stop the spread of COVID-19”

Response: thank you very much for your comment. We have included further literatures based on your recommendation in the revised version of the manuscript. 

Question #10) Line 127, what is tottery?

Response: sorry for the editorial problem we have created. Hence, we have edited the word tottery to the word lottery method in the revised manuscript. 

Question #11) Was distance to water sources measured with a tape measure?

Response: If the water source is available in the premise of the Healthcare facilities, no need of measuring the distance. On the other hand, when the water source is out of the premise, the distance of water source was measured using appropriate distance measuring tool.

Question 12). Statistics and other analyses are not performed to a high technical standard and are not described in sufficient detail. This is listed as a limitation later, but needs to be done prior to acceptance of the paper.

Response: thank you very much for your comment. The statistical analysis depends on the objective of the study. The objective of the current study can be expressed by using descriptive statistics using frequency, mean with standard deviation. Furthermore, we have presented the absence of advanced statistical analysis as the limitation of the study. 

Question#13). The N=70, there is not a large enough dataset to have results to 0.1%. Please round all results presented to the nearest whole number.

Response: We have incorporated your comment in the revised version of the manuscript.

Question#14) The study needs to compare to the recent Ethiopia WASH in healthcare facility work during COVID-19, this is a major gap of this manuscript https://washdata.org/sites/default/files/2020-12/WHO-UNICEF-2020-wash-in-hcf.pdf

Response : Thank you very for your comment we have tried to use further papers which were conducted indifferent parts of the world for comparison with the finding the current study. (See the revised version of the manuscript) 

Question#15.) The study presents the results of original research, but does not compare to other recent studies in Ethiopia or other low- and middle-income countries during COVID-19.

Response Thank you very for your comment. We have used similar papers which were conducted in low resource setting mainly in African countries in order to compare with this study finding. But there is limitation of the stud in these areas (See the revised version of the manuscript)

Question#16.) Conclusions presented are not supported by the data. Expand the conclusions. How could the findings of this work be implemented for better access, not just in Ethiopia but lessons learned for other low- and middle-income countries? At least give some practical options for solutions in a bulleted list. What can be done within resource limited environments? How much financial investment is needed at a minimum to improve conditions? Also, I missed seeing what further research you can suggest in this field.

Response: We have tried to modify the conclusion based on the finding of the study. (See the revised version of the manuscript).

Question#17.) Finally, when you submit the corrected version, please do check thoroughly, in order to avoid grammar, syntax or structure/presentation flaws - please seek for professional English proofreading services or ask a native English-speaking colleague of yours in order to refine and improve the English in your paper.

Response: Thank you very much for your valuable comment. We have modified all the problems you mentioned. (See the revised version of the manuscript) 

 Response to Reviewer 2

Question #1 Healthcare facilities should not be in capital in the title, or throughout the manuscript. Only proper nouns should be in capital letters. 

Response: thank you for your comments; hence we have incorporated it in the revised manuscripts 

Question #2 sampling why not you use proportional allocation to the number of health care facilities since all health facilities are known by the regional health bureau. 

Response: Ok thank you for your comments. We have lists of Healthcare facilities in the study catchment. Therefore we can use sample random sampling technique to select the study participants using the lists of the healthcare facilities as the sampling frame.

Question #3 Line 33 add the word “by” before the word “using “ and add “interviewer administered” before “structured questioner” 

Response: we have amended it accordingly. 

Question #4 Line 50 who are nongovernmental health facilities? Change the word either NGO or partners

Response: sorry for the confusion. We want to say concerned partners. Hence we have amended it accordingly.

Question #5 line 98 it doesn’t state in which country specifically that CoVID-19 is occurred on March 13, Hence re-write this sentence.

Response: sorry for the confusion; it is to mean for Ethiopia therefore, we have amended it accordingly in the revised manuscript.

Question #6 Line99-100 the sentence is not complete.

Response: sorry for the confusion, we have improved it in the revised manuscript. 

Question #7 Line 101 278 HCFs are in nationwide, or regionally or zonal level???????

Response: sorry for the confusion, the report was representing the nationwide of Ethiopia. Hence we have incorporated in the revised version.

Question #8 Remove line 113,114,115 which talks about the boarder of Wollo zone that much is not necessary

Response: thank you for your important comment. We have removed in the revised manuscript

Question #9 Line 121 All HCFs which is present…… change the word present by the word “exist”

Response: thank you for your comment we have modified it accordingly.

Question #10 Line 125 why not you mention the number of the total health facilities found in the zone to see whether the sample (70) is representative or not? And what was your base to take 70? Why not 80, 90,100 or above? Out of the total sample, 80% are private clinics what was your base to take larger sample from private clinics.

Response: thank you for your comment we have already mentioned the total number of healthcare facilities in the method section of the manuscript. The sample was taken based on the number of existence healthcare facilities in the study area. The number of clinics in the study area was higher than other healthcare facilities which motivate us to take larger sample than others. The base for the sample was resource limitations and lockdown which restricts the movements of data collectors. 

Question #11 Line 125 Since there is enough data on the number of health institutions found in the zonal health department, why don’t you use proportional sampling technique rather than simple random to make it more uniform and representative of each health facility category?

Response: thank you for your comment. We have frames of healthcare facilities in the health authorities of the study area. Hence, we used simple random sampling techniques using the lottery method to avoid bias in the selection of the study participants. 

Question #12 Line 127-131 how qualitative data were collected? Is it in-depth interview or FGDs? Explain well which method is best in exploring the challenges of an issue?

Response: the qualitative data was collected using in-depth interview. But to explore the challenges of such types of issues FGD may be more appropriate

Question #13 Line 150 what about the definition of limited access for sanitation??

Response: sorry for the confusion we have incorporated it in the revised manuscript.

Question #14 Line 165 contextualized to the study……

Response: we have corrected it accordingly in the revised manuscript.

Question #15 Line 170 you use spot observation to collect data but you didn’t mention the result that you obtain through this method in your result part.

Response: sorry for the confusion we have created; as you see in the method section we used different methods of data collection which is interview and observation. The data which was collected by these methods are presented in the result section of the manuscript. For example the data of handwashing facilities were collected by observational methods

Question #16 Line 188 how the qualitative data were analyzed? It is not exhaustively written. Hence you should mention the method that used to analyze it.

Response: the qualitative data was analyzed using thematization method.

Question #17 Line 204-206; do you think that your result represent the HCFs that are found in the zone?

Response: yes; in the study catchment there are heath posts, clinics, health centers, and hospitals. We tried to take samples from different categories of healthcare facility categories based on the number of healthcare facilities existed during the data collection time.

Question #18 Line 206 are an environmental health professionals the only professionals who coordinates the WaSH activities? Did you ask the presence of even other professionals in the position?

Response: Environmental Health professionals are not the only professional who coordinates WASH activities but they are the best professionals who can coordinate such types of activities. Furthermore, we have assessed as there are there professionals who are wring on the position. 

Question #19 Line 207 average is not the appropriate measures rather you use mean +/- SD since the average is 55 but the highest value is 320, do you see the gap?

Response: thank you for your comment. Despite we used mean and the highest value in the result section, we have expressed it using mean with standard deviation in the tables. Therefore, we have incorporated it in the result section of the revised manuscript. 

Question #20 Line 208; though the highest number of patient flow is in the hospital, you included in your study only 7% of them. Do you think your sample is representative?

 Response: we think the sample is representative of the whole sample. This is because the existed number of hospitals in the study catchment is less than 5% but we take larger samples which are 7% to accommodate the highest client flow rate in such types of healthcare facilities. 

Question #21 Line 234 I think it is better to write “…HCFs was unbalanced patient load with functional…. ”

Response: Thank you for the comment. We have amended it accordingly in the revised manuscript.

Question #22 Line 240 it is better to change the word “the floor” by the word “the slab”…..disposal of “faces” by “faeces “ after that add the word “and” before urine

Response: we have incorporated your comments accordingly in the revised manuscripts.

Question #23 Line 242 ….challenges of hygienic facilities

Response: We have modified the comment accordingly in the revised manuscript. 

Question #24 Line 243 since it is 35 or half of your sample, you can’t say more than half of…..

Response: sorry for the problem. Hence, we have corrected it accordingly in the revised manuscript.

Question #25 Line 248 “proper waste management” what does it mean in this research context. It needs operational definition for this research

Response: thank you for the comment. We have incorporated the comment in the revised manuscript.

Question #26 Line 248 …..and they disposed off …. Add one “f”

Response: we have corrected it accordingly in the revised manuscript. 

Question #27 Lines 249-252; the sentence is not clear. Is it about cleaning or cleaning protocol or the janitors training status?

Response: sorry for the confusion. We re-write it in the revised manuscript. 

Question #28 Line 253-255 I didn’t understand about your sample the person from private health care facilities customer? If it is so remove it since they are two independent setting i.e governmental and private hospitals you compared two different institutions. Besides, in your sampling section, there is no list of participant from private health institution users. From where did you bring this information?

Response: the study incorporates both government and non-governmental health care facilities (health center, clinics, and hospitals) which exist in the study setting. In the qualitative section, we have selected individual participants using purposive sampling technique using an in-depth interview from both types of institutions to assess the major challenges faced in there heath institutions mainly after the occurrences of COVID-19 pandemic. Therefore, the finding of the current study represents governmental and non- governmental healthcare facilities of the study catchment. 

Question #29 Line 262-263 it is better to add references to the written sentence

Response: thank you for your comment. We have added reference accordingly in the revised manuscript. 

Question #30 Line 265-267 the sentence which describes the aim should be omitted since it is already mentioned earlier

Response: Ok; we have omitted in the revised manuscript.

Question #31 Line 267. Can we say all HCFs have accessed to water supply? In you result part line 214, 11% of them have no water during data collection time. Some time it is more than that. Hence you should modify this statement

Response: thank you for the comment. We have amended it accordingly (see the revised manuscript)

Question #32 Line 267-271 you should treat each component independently otherwise there will be writing the result repeatedly which will be boring to the reader. Hence write the discussion separately for water, sanitation and hygiene 

Response: thank you; we have modified it in the revised manuscript. Hence write the discussion separately for water, sanitation and hygiene in the revised manuscript.

Question #33 Line 273 and 274 it needs re-writing and is it possible to put the different country status by average number? Why not you mention for each country status to know and compare it 

Response: we have modified it accordingly in the revised manuscript.

Question #34 Line 275; your justification doesn’t convince the reader i.e why don’t you find the similar setting to compare it either rural or urban?

Response: we have tried to search out similar setting for easy comparison of the current finding with other finding. But, we cannot get sufficient literatures based on your recommendation

Question #35 Line 278 your justification is the variation in the HCFs why not you compare with similar setting? 

Response: we have tried to search out similar setting for easy comparison of the current finding with other finding. But, we cannot get sufficient literatures based on your recommendation

 Question #36 Line 278 remove the bracket at the end of line 278

Response: we have removed it in the revised manuscript. 

Question #37 Line 279-282 “almost all …….in private clinics” this statement should be placed after discussing water, sanitation and hygiene results and even compare the result and the situation with studies done in other parts of Ethiopia.

Response: thank you for your comment; we tried to compare this study finding with other countries mainly in low resource setting for comparison. But there is deficiency of study conducted in Ethiopia in this aspect. 

Question #38 Line 287 you said that the ratio 11.5 is greater than WHO standard. What is the WHO standard put the number. Besides, there is no reference, hence put your reference

Response: we have incorporated your comment in the revised manuscript.

Question #39 Line 288 the statement “….between the level….” Should be changed to between the HCFs. Even it is better to put the ratio result (in number) with in each HCF.

Response: Ok, we have incorporated your comment in the revised manuscript. 

Question #40 Line 291-301 the paragraph seems the result part since you didn’t discuss any of your result by comparing your result with the other studies. Hence this paragraph needs re-writing again

Response: thank you for your comment. Hence, we have tried to compare the current finding with other studies conducted in different parts of the word, mainly in developing countries.

Question #41 Line 304 “HCFs had no function…..” change the word function to “functional”

Response: Ok; we have modified in the revised manuscript. 

Question #42 Line 305 what about the other study setting you didn’t discuss your finding simply put the number that you wrote in the result part here again.

Response: thank you for your comment. Hence, we have tried to modify the discussion by comparing this finding with other study finding. 

Question #43 Line 307-309; this justification is for the 60% (HCFS had no function hand washing facilities) or 21% (HCFs had functional handwashing facilities with water and soaps)? It is confusing discussion part. Hence it is better to treat independently.

Response: sorry for the confusion we have created. Therefore, we have amended it in the revised manuscript. 

Question #44 Line 314-320 you simply put your result again that you wrote in your result part. Hence it needs further discussion by comparing your result with other findings and give your justification why the variation occurs.

Where is the qualitative data discussion part? you didn’t put anything. Why you collect the qualitative data? And how you analyzed it? Is it by open code, thematic, ATLAS-Ti or by what technique? Nothing is said about this. Please write something on it. You said in your methodology mixed where is the mixed nature? I didn’t see it or realized it.

Response: thank you very much for your critical comment. We have incorporated the qualitative section in discussion in line with the quantitative one. The qualitative data are very important in identifying the major challenges of WASH facilities in the healthcare setting. The qualitative data was analyzed using the thematization technique. The mixed nature of the study incorporates both qualitative and quantitative. The access of WASH facilities were studied using quantitative method whereas the challenges of WASH facilities were studied using the qualitative technique 

Question #45 Line 323 “exclusion of health post” I preferred you were included these health post rather than the private clinics that you included. The reason is that 

1. There is a mix up of different setting that is governmental and private which are almost completely different setting in Ethiopian condition

2. Your study design is not comparative cross sectional method 

3. You didn’t compare the result of private clinics with the government or even to the other similar setting of different countries. The data that you got from private clinics are confined in the governmental HCFs and treated as they are governmental HCFs

 Response: The study tried to associate the COVID-19 prevention in healthcare setting. in the case of Ethiopia particularly in the study setting, most of the populations usually get treatment for different healthcare facilities including hospitals, heath centers, and clinics. In the case of heath posts, the major service usually focuses on the prevention of disease including the current pandemic of COVID-19. Hence, clients are not expected to visit the health post which motivates us to exclude from the study. 

Question #46 Limitation part what about the cross-sectional nature of the data? 

Response: We have incorporated it in the revised manuscripts accordingly. 

Question #47 Line 335 make correct on the word “…..concerned exerts……” change to concerned experts

Response: we have corrected it in the revised manuscript. 

Question #48 Annex Table, 3 on page 24 “overall latrine cleanliness” i.e high, medium, low, it needs operational definition.

Response: thank you for your comment. We have included the operational definition of the word latrine cleanliness in the revised manuscript.

---

## [Decision Letter · Decision Letter 1]

28 Feb 2022

PONE-D-21-35346R1Access to and challenges in water, sanitation, and hygiene in healthcare facilities of Northeastern Ethiopia in the COVID-19 era: A mixed methods evaluationPLOS ONE

Dear Dr. Berihun,

Thank you for submitting your manuscript to PLOS ONE. After careful consideration, we feel that it has merit but does not fully meet PLOS ONE’s publication criteria as it currently stands. Therefore, we invite you to submit a revised version of the manuscript that addresses the points raised during the review process. Many thanks for the improvements to the manuscript but there is still some work to be done to bring it up to the required standard.  The findings need to be compared to other studies.   These can be within Sub Saharan Africa but must be outside Ethiopia.   Searching in a free engine like Google Scholar should bring the desired results.   Please look carefully at the other points from reviewer 1 too.

We look forward to receiving your revised manuscript.

Kind regards,

Alison Parker

Academic Editor

PLOS ONE

Reviewers' comments:

Reviewer's Responses to Questions

**Comments to the Author**

1. If the authors have adequately addressed your comments raised in a previous round of review and you feel that this manuscript is now acceptable for publication, you may indicate that here to bypass the “Comments to the Author” section, enter your conflict of interest statement in the “Confidential to Editor” section, and submit your "Accept" recommendation.

Reviewer #1: (No Response)

Reviewer #2: All comments have been addressed

2. Is the manuscript technically sound, and do the data support the conclusions?

Reviewer #1: Partly

Reviewer #2: Yes

3. Has the statistical analysis been performed appropriately and rigorously? 

Reviewer #1: No

Reviewer #2: Yes

4. Have the authors made all data underlying the findings in their manuscript fully available?

Reviewer #1: Yes

Reviewer #2: Yes

5. Is the manuscript presented in an intelligible fashion and written in standard English?

Reviewer #1: No

Reviewer #2: Yes

6. Review Comments to the Author

Reviewer #1: 1.) I continue to be concerned with the wider use of literature. For the use of “worldmeter” references 18 and 22, I would prefer to see official Ethiopian government sources. Also, the use of a WHO/UNICEF 2015 JMP report in reference 38. Please update and use the most recent 2021 WHO/UNICEF report.

2.) The study presents the results of original research, but does not compare to other recent studies in Ethiopia or other low- and middle-income countries during COVID-19.

3.) As previously commented, the authors have not made this change. “The N=70, there is not a large enough dataset to have results to 0.1%. Please round all results presented to the nearest whole number.” In Revision 1, Table 1 and 2 still list results to 0.1%

4.) Finally, when you submit the corrected version, please do check thoroughly, in order to avoid grammar, syntax or structure/presentation flaws - please seek for professional English proofreading services or ask a native English-speaking colleague of yours in order to refine and improve the English in your paper.

Reviewer #2: all comments that I raised in the first round were properly addressed by the authors. That was nice. But I have one comment on the reference section. Your references are not similar in style. Hence you should rewrite all references again

7. PLOS authors have the option to publish the peer review history of their article (what does this mean?). If published, this will include your full peer review and any attached files.

Reviewer #1: No

Reviewer #2: No

---

## [Author Response · Author response to Decision Letter 1]

9 Mar 2022

Response to Reviewer 1

 #1: I continue to be concerned with the wider use of literature. For the use of “Worldometer” references 18 and 22, I would prefer to see official Ethiopian government sources. Also, the use of a WHO/UNICEF 2015 JMP report in reference 38. Please update and use the most recent 2021 WHO/UNICEF report. 

Response:-thank you very much for your comments. Hence, we have incorporated your comments in the revised version of the manuscript.

2.) The study presents the results of original research, but does not compare to other recent studies in Ethiopia or other low- and middle-income countries during COVID-19.

Response:-thank you for your constrictive comment too. We have tried to use researches which were conducted in Ethiopia and other low and middle countries as much as possible in the revised manuscript.

3.) As previously commented, the authors have not made this change. “The N=70, there is not a large enough dataset to have results to 0.1%. Please round all results presented to the nearest whole number.” In Revision 1, Table 1 and 2 still list results to 0.1%

Response:-ok we have modified it based on your comments in the revised manuscript (see the revised version).

4.) Finally, when you submit the corrected version, please do check thoroughly, in order to avoid grammar, syntax or structure/presentation flaws - please seek for professional English proofreading services or ask a native English-speaking colleague of yours in order to refine and improve the English in your paper.

Response:-thank you for the comment; we have tried to incorporate all issues you have raised.(see the revised version of the manuscript)

Response to reviewer 2

#1: all comments that I raised in the first round were properly addressed by the authors. That was nice. But I have one comment on the reference section. Your references are not similar in style. Hence you should rewrite all references again

Response:-thank you for the comment. We have modified the references based on the standard guideline of the journal. (see the revised version of the manuscript)

---

## [Editor Report · Decision Letter 2]

31 Mar 2022

PONE-D-21-35346R2Access to and challenges in water, sanitation, and hygiene (WASH) in healthcare facilities of Northeastern Ethiopia in the COVID-19 era: A mixed-methods evaluationPLOS ONE

Dear Dr. Berihun,

Thank you for submitting your manuscript to PLOS ONE. After careful consideration, we feel that it has merit but does not fully meet PLOS ONE’s publication criteria as it currently stands. Therefore, we invite you to submit a revised version of the manuscript that addresses the points raised during the review process.

I have assessed your submission, and whilst the science is now adequate, I have concerns about the overall readability of the manuscript. I therefore request that you revise the text to fix the grammatical errors and improve the overall readability of the text. 

PLOS ONE suggests you have a fluent, preferably native, English-language speaker thoroughly copy-edit your manuscript for language usage, spelling, and grammar. If you do not know anyone who can do this, you may wish to consider employing a professional scientific editing service. Whilst you may use any professional scientific editing service of your choice, PLOS has partnered with both American Journal Experts (AJE) and Editage to provide discounted services to PLOS authors. Both organizations have experience helping authors meet PLOS guidelines and can provide language editing, translation, manuscript formatting, and figure formatting to ensure your manuscript meets our submission guidelines. To take advantage of the partnership with AJE, visit the AJE website (https://www.aje.com/go/plos/) for a 15% discount off AJE services. To take advantage of the partnership with Editage, visit the Editage website (www.editage.com) and enter referral code PLOSEDIT for a 15% discount off Editage services. 

Upon re-submission, please provide the following: 

Please note that PLOS ONE does not copy-edit accepted manuscripts and that one of the criteria for publication is that articles must be presented in an intelligible fashion and written in clear, correct, and unambiguous English (http://www.plosone.org/static/publication#language). If the language is not sufficiently improved, I may have no choice but to reject the manuscript.

We look forward to receiving your revised manuscript.

Kind regards,

Alison Parker

Academic Editor

PLOS ONE
---

## [Author Response · Author response to Decision Letter 2]

23 Apr 2022

Question #1 I have assessed your submission, and whilst the science is now adequate, I have concerns about the overall readability of the manuscript. I therefore request that you revise the text to fix the grammatical errors and improve the overall readability of the text. 

Response: Thank you for this remark. We have revised the whole manuscript to reduce the grammatical errors and improve the overall readability of the text by the help of known researcher Dr Metadel Adane Mesifin (Please see the revised version for each section).

Question #2 Please review your reference list to ensure that it is complete and correct. If you have cited papers that have been retracted, please include the rationale for doing so in the manuscript text, or remove these references and replace them with relevant current references. Any changes to the reference list should be mentioned in the rebuttal letter that accompanies your revised manuscript. If you need to cite a retracted article, indicate the article’s retracted status in the References list and also include a citation and full reference for the retraction notice.

Response: thank you very much for critical comment. Hence we have improved the whole listed references using the guideline of PLOSE ONE (see the revised version for each section).

---

## [Editor Report · Decision Letter 3]

27 Apr 2022

Access to and challenges in water, sanitation, and hygiene (WASH) in healthcare facilities of Northeastern Ethiopia in the early phase of the COVID-19 pandemic: A mixed-methods evaluation

PONE-D-21-35346R3

Dear Dr. Berihun,

We’re pleased to inform you that your manuscript has been judged scientifically suitable for publication and will be formally accepted for publication once it meets all outstanding technical requirements.

Kind regards,

Alison Parker

Academic Editor

PLOS ONE
---

## [Editor Report · Acceptance letter]

4 May 2022

PONE-D-21-35346R3 

Access to and challenges in water, sanitation, and hygiene in healthcare facilities during the early phase of the COVID-19 pandemic in Ethiopia: A mixed-methods evaluation 

Dear Dr. Berihun:

I'm pleased to inform you that your manuscript has been deemed suitable for publication in PLOS ONE. Congratulations! Your manuscript is now with our production department. 

Kind regards, 

on behalf of

Dr. Alison Parker 

Academic Editor

PLOS ONE